# The psychosocial burden of women seeking treatment for breast and cervical cancers in Ghana's major cancer hospitals

Patience Gyamenah Okyere Asante[1], Adobea Yaa Owusu[1], Joseph Ransford Oppong[2], Kingsley E. Amegah[3], Edward Nketiah-Amponsah[4]*

1 Institute of Statistical, Social and Economic Research, University of Ghana, Legon, Accra, Ghana,
2 Department of Geography, University of North Texas, Denton, Texas, United States of America,
3 Department of Data Science and Economic Policy, School of Economics, University of Cape Coast, Cape Coast, Ghana, 4 Department of Economics, University of Ghana, Legon, Accra, Ghana

* enamponsah@ug.edu.gh

## Abstract

### Background

Breast and cervical cancers remain the most common cancers and the leading cause of cancer deaths in Ghana. Non-communicable diseases such as cancers, have been associated with psychological burdens such as anxiety and depression disorders as well as severe mental disorders such as bipolar disorder. As such the World Health Organisation has noted that mental health and well-being are crucial in reducing the NCD burden.

### Methods

A convergent mixed method approach was used to ascertain the psychosocial burden of breast and cervical cancer patients who sought treatment in three major cancer hospitals in Ghana. Primary data were collected using a questionnaire and an interview guide from 298 breast and cervical cancer patients seeking treatment at the Korle-Bu and Komfo Anokye Teaching Hospitals as well as the Sweden Ghana Medical Centre. Qualitative analysis was done using thematic content analysis while quantitative analysis was done using logistic regression.

### Results

The findings of the study showed that patients not only battled with psychological burdens such as anxiety, depression, pain, stigma, fear of death and loss of spouses but also struggled with physical, social, and dietary restrictions. Patients with low educational levels and income status, retired or unemployed, and/or had larger household sizes suffered more psychosocial burdens.

### Conclusion

There is a need for liaison psychiatrists and health psychologists to assist oncologists to provide psychological support such as free and routine counselling services for cancer patients

**Data Availability Statement:** The data for this paper are available upon reasonable request from Dr. Patience Gyamenah Okyere, upon a formal request. There are ethical and confidential concerns

about breast cancer within the Ghanaian cultural setting. The patients were promised that their responses will not be shared with the public, unless it becomes necessary to share it with persons who may request to use it privately—or to Editors of publishing outlets, when some of them showed concerns during the interviews about keeping the data confidential. Questions on the data can be directed at Dr. Patience Gyamenah Okyere Asante (Institute of Statistical, Social and Economic Research. University of Ghana, Legon. Accra, Ghana; email: patg2014@gmail.com) or at the Coordinator of the ISSER PhD program at the following e-mail address: (isser-graduate@ug.edu.gh).

**Funding:** The authors received no specific funding for this work

**Competing interests:** The authors have declared that no competing interests exist.

and their caregivers. Educational campaigns on mainstream and social media need to be intensified to demystify the stigma surrounding cancers in Ghana.

## 1.0 Introduction

The global rising burden of non-communicable diseases (NCDs) especially among women, has become a considerable health and development issue. These pose a challenge to the achievement of SDG goals 3 and 5 [1]. Non-communicable diseases such as cancers, diabetes and cardiovascular diseases have been associated with psychological burdens such as anxiety and depression disorders as well as severe mental disorders such as bipolar disorder [2–4]. As such WHO has noted that mental health and wellbeing are crucial in reducing NCDs burden [3].

Female breast cancer was the most commonly diagnosed cancer with an incidence of 2.26 million cases in 2020 [5, 6]. Breast cancer was also the leading cause of cancer death in twelve regions of the world while cervical cancer was the leading cause of death in three regions in Sub-Saharan Africa including Eastern Africa [5, 6]. In Ghana, breast and cervical cancers remain the most common female cancers and the leading cause of cancer deaths. In 2018, 22,823 cancer cases were recorded in Ghana with breast and cervical cancers accounting for 20.4% and 13.5%, respectively [7]. In 2020, the number of new cancer cases in Ghana increased to 24,009 with breast cancer remaining the most commonly reported, while lung cancer overtook cervical cancer as the second highest [8].

Apart from the huge economic cost that cancer treatment poses to patients and their families, they also bear high psychosocial burdens. For instance, studies have shown that people with cancer experience depressive spectrum disorders, anxiety disorders and post-traumatic stress disorder [9, 10]. About 60% of cancer patients suffer from depressive spectrum disorders such as major, minor, and persistent depression and other forms of depression such as demoralization [9]. Even though these psychological burdens have negative impacts on cancer treatment, recovery, quality of life, and survival of cancer patients, they have mostly been given little recognition and treatment [9, 10].

While considerable work has been done on the increasing incidence and prevalence of cancer diseases in Ghana, there is limited literature on the psychosocial experiences of cancer patients in the Ghanaian context. Also, studies on the cancer burden on patients have focused on the economic aspects to the neglect of the psychosocial burden patients bear due to their illness. This paper examines the psychosocial experiences of breast and cervical cancer patients who sought treatment in the three major cancer hospitals in Ghana. The goal of this paper is to shed light on the psychosocial burdens that breast and cervical cancer patients bear which cannot be easily quantified in economic terms.

## 2.0 Methodology

### 2.1 Study setting

Cancer care in Ghana is centralized with two major teaching hospitals resourced with oncology units or directorates to provide cancer care for patients. These hospitals namely, the Korle-Bu and Komfo Anokye Teaching Hospitals located in Accra and Kumasi and are the first and second largest teaching hospitals in Ghana and serve as referral centres for cancer cases diagnosed across the country. Apart from these two major public hospitals, there are a few private hospitals which provide private cancer care in Ghana, notable among them is the

Sweden Ghana Medical Centre. These three hospitals were purposively selected as sites for the study as most cancer patients seek treatment from these hospitals.

## 2.2 Selection of study participants

The study population consist of breast and cervical cancer patients who were seeking treatment at the 3 major cancer hospitals in Ghana namely the Korle-Bu and Komfo Anokye Teaching Hospitals and the Sweden Ghana Medical Centre. Since data on breast and cervical cancer prevalence in Ghana are not readily available making sample size computations impossible or difficult, patients were purposively selected from the various hospitals based on the following inclusion criteria:

a. Patients above 18 years

b. Patients who are clinically diagnosed with breast or cervical cancer.

c. Patients who have started biomedical treatment for 6 months or more.

Patients who met the inclusion criteria were approached for informed consent after which they were recruited to participate in the study.

Two hundred and ninety-eight (298) cancer patients took part in the study, forty-five (45) respondents were followed up for qualitative interviews after they had participated in the initial survey. Qualitative interviews for patients at the various hospitals were halted once saturation points were reached and new interviews did not provide new insights into the phenomena being investigated.

## 2.3 Data collection instruments and method

A researcher-administered questionnaire was used to collect quantitative data while an interview guide was used to collect qualitative data. Data collection took place between the months of March and August 2020. With an introduction from oncology nurses, and after informed consent was sought, patients were engaged at the treatment facilities while they waited to be attended to. Patients responded "yes" or "no" to a series of questions on their psychosocial health. The variables covered in the study include anxiety, depression, psychosocial distress, physical pain, financial-related stress, social restrictions, dietary restrictions, stigma, and disclosure of cancer diagnosis. Patients who agreed to follow-up sessions were approached for face-to-face interview sessions where they were asked to share their experiences in relation to the variables listed above. Interviews were conducted in English, or Twi (a local dialect) based on patients' preference and were audio recorded with patients' consent.

## 2.4 Data analysis

The quantitative data were analysed using Microsoft Excel 360 and STATA 16.1. Frequency tables yielded the percentages of patients who experienced the various dimensions of psychosocial burden. Additionally, a multivariate logistic regression analysis was conducted (using patients' socio-demographic characteristics as independent variables and selected measures of psychosocial burden as dependent variables) to explore which patients were more prone to experience psychosocial burden. The qualitative data were analysed using thematic content analysis. Pre-set themes had been generated ahead of time based on the research objectives which guided the questions for the face-to-face interviews. The recorded interviews were transcribed verbatim, and the data was condensed into a few manageable groups, tables, and transcripts with specific ID numbers assigned to each patient. First and second-cycle codes were generated according to the pre-set themes. Broader themes were then generated from the pre-

set themes to guide the discussion of findings making room for new themes which emerged from the interviews.

## 3.0 Findings and results

Patients reported psychological burdens resulting from anxiety, depression, fear, and financial-related stress, and stigma. They also reported that they battled with physical, social, and dietary restrictions.

### 3.1 Anxiety and financial-related stress

Almost all the patients in the study had anxieties about their general health and the outcome of their treatment. They battled with financial-related stress as they worried about getting money for their treatment. Patients also worried about how to meet the needs of their dependents. Most patients worried about the burden they posed to their family members and loved ones. The quantitative analysis stress (Table 1) showed that 91.4% and 71.0% of breast cancer patients in public and private hospitals, respectively, reported anxiety. Eighty-six percent (86%) of cervical cancer patients also reported being worried because of their illness. About 90.9% and 19.4% of breast cancer patients in the public and private hospitals in the study reported financial-related stress while 92.6% of cervical cancer patients experienced financial related. Majority of breast cancer patients who sought treatment at the private hospital, mostly teachers, received comprehensive health insurance from the Ghana National Association of Teachers (GNAT) cancer fund which reduced their financial stress.

With regard to worry, a 45-year-old cervical cancer patient stated:

*"I get very worried; nobody likes to be sick. Your life gets disrupted. I am worried that I cannot care for my family the way I want to. I worry about the stress my husband is going through."*

Regarding financial-related stress, a 55-year-old cervical cancer patient lamented:

**Table 1. Psychosocial burden of breast and cervical cancer treatment in public and private hospitals.**

|  | Public Hospital | | Private Hospital |
| --- | --- | --- | --- |
| **Statements** | **Breast** | **Cervical** | **Breast** |
| **Anxiety, depression, and psychological distress** |  |  |  |
| I often get worried and depressed because of my illness | 91.4 | 86.4 | 71.0 |
| I experience financial-related stress | 90.9 | 92.6 | 19.4 |
| There is stress on other members of my household, and I am worried about it | 83.3 | 91.4 | 48.4 |
| **Fear of death and loss of loved ones** |  |  |  |
| I am afraid that I may lose my husband (based on married respondents only) | 80.1 | 85.2 | 64.5 |
| I am sometimes afraid that I will die | 87.6 | 82.7 | 77.4 |
| **Social, physical, and dietary restrictions** |  |  |  |
| I am unable to attend social events as I used to | 81.7 | 82.7 | 74.2 |
| I must rest more, I feel restricted | 88.2 | 85.2 | 83.9 |
| I must change the food I eat; I feel restricted | 70.4 | 72.8 | 83.9 |
| I go through much physical pain | 80.7 | 93.8 | 83.9 |
| **Stigma and concealment of cancer diagnosis** |  |  |  |
| I struggle with perceived stigma | 42.5 | 60.5 | 48.4 |
| I must try hard to hide my illness from people | 80.1 | 77.8 | 77.4 |

Source: Authors' data

*"I get very anxious and depressed anytime my appointment date is getting close. How to get money for treatment is always a challenge. When you see that I am sitting quietly, it means I am in deep thought. Worry alone can even kill you. The cost of treatment is too huge for us."*

A 44-year-old breast cancer patient also bemoaned:

*"It is the financial stress that is killing us. . .you see; I worry so much about how to get money for the treatment. When you worry, you can't eat, you can't sleep! You simply get depressed.*

With reference to emotional pain, A 47-year-old cervical cancer patient also added:

*"I cry a lot because of this illness and the associated cost. The nurses know me as the crying patient. When they mention the amount of money I must pay, I just burst into tears. I do not know where this illness came from. It has brought so much financial distress to my family. I have become a beggar, always asking for money for my treatment."*

Even though most of the patients knew that worry was not good for their health, they could simply not avoid it because of the financial burden they faced and the consequences of not being able to raise money for treatment. Even though relatives and oncology staff encouraged patients to desist from worrying, it was not enough to allay their anxieties as their challenges persisted.

A multivariate logistic regression (Table 2) revealed a statistically significant association between the type of cancer, patient's highest level of education and worrying among patients due to their illness. Cervical cancer patients had significantly lower odds [OR = 0.29 (95% CI; 0.11, 0.65), p = 0.003] of worrying due to their illness compared with patients with breast cancer. Respondents with secondary [OR = 0.19 (95% CI; 0.04, 0.99), p = 0.048] and tertiary [OR = 0.05 (95% CI; 0.01, 0.29), p = 0.001] levels of education were less likely to worry about their illness compared with those with no formal education. Patients who were highly educated and could read about cancer experienced less anxiety about dying from their illness compared to patients who could not read and had little information about cancer.

### 3.2 Fear of death, loss of spouses and care for young dependents

Patients also battled with fear of the unknown as they were not sure of the outcome of their treatment. This was true especially for those who had financial challenges as they were afraid that they might die if they did not get money to complete their treatment on time. Other patients had fears about losing their spouses to divorce or separation due to the high financial burden on their spouses, stemming from treating their cancer. Some patients reported that their spouses gave up on them or threatened to do so if their illness worsened.

Most (87.6% and 77.4%) of breast cancer patients in the public and private hospitals studied, respectively, and 82.7% of the cervical cancer patients studied had fears that they would die because of their illness. The majority, 80.1% of breast cancer patients in the public hospitals compared to 64.5% in the private hospital feared losing their husbands to divorce or separation because of their illness. Similarly, 85.2% of cervical cancer patients had fears of losing their husbands to divorce or separation due to their illness (Table 1). A 48-year-old cervical cancer patient said her greatest fear was who would take care of her children should she die. Her fear was not of death but the upkeep and well-being of her young dependents. She explained:

*"I am not necessarily afraid of death; my fears are about who will take care of my children when I am no more. It is my biggest fear."*

**Table 2. Determinants of worrying among patients due to illness among respondents.**

| Variable | OR [95% CI] | P-value |
|---|---|---|
| Occupation | | |
| Waged worker | 1.00 | |
| Self-Employed | 0.73 [0.25, 2.13] | 0.566 |
| Unemployed | 0.62 [0.06, 6.67] | 0.691 |
| Retired | 0.57 [0.18, 1.81] | 0.337 |
| *Monthly Income (GHC/USD) | | |
| Less than GHC 500 ($89.2) | 1.00 | |
| GHC 500–999 ($89.2–178.4) | 0.71 [0.25, 2.03] | 0.523 |
| GHC 1000–1999 ($178.6–356.9) | 1.43 [0.45, 4.48] | 0.543 |
| GHC 2000–2999 ($357.1–535.5) | 0.80 [0.23, 2.82] | 0.727 |
| GHC 3000–3999 ($535.7–714.1) | 0.90 [0.13, 6.24] | 0.913 |
| GHC 4000+ ($714.3) | 0.49 [0.07, 3.29] | 0.459 |
| Stage of cancer on diagnosis | | |
| Early stage | 1.00 | |
| Middle stage | 1.25 [0.55, 2.82] | 0.591 |
| Advanced stage | 1.62 [0.16, 6.17] | 0.679 |
| Residential Status | | |
| Rural | 1.00 | |
| Urban | 0.86 [0.35, 2.13] | 0.745 |
| Type of Hospital | | |
| Public | 1.00 | |
| Private | 3.62 [0.68, 9.21] | 0.131 |
| Type of cancer | | |
| Breast cancer | 1.00 | |
| Cervical cancer | 0.27 [0.11, 0.65] | **0.003** |
| Highest level of education | | |
| No formal education | 1,00 | |
| Primary | 0.29 [0.05, 1.61] | 0.157 |
| Secondary | 0.19 [0.04, 0.99] | **0.048** |
| Tertiary | 0.05 [0.01, 0.29] | **0.001** |

OR = Odds Ratio, CI = Confidence Interval.

*(Inter-Bank Exchange Rate-Month Average, GHC5.6 = 1 USD, June 2020 (when data was collected)

Source: Authors' data, 2022

A 51-year-old breast cancer patient also stated:

*"I am sometimes afraid that I may die because you see people who die because of their illness."*

A 34-year-old cervical cancer patient recalled:

*"A lot of the people I started the treatment with have died. Sometimes it really gets to you, and you get scared that you might not live long. If you don't have strong support from your family, you will die. Even the side effect of the treatment alone can kill you."*

A 41-year-old breast cancer patient also lamented:

*"I am afraid that I may lose my husband because his attitude has changed due to the huge financial burden."*

The fear of death among the respondents was significantly associated with the stage of cancer, the type of health facility the patient received care at, and the highest level of education attained by the patient. Patients who were at the advanced stage of cancer were more likely [OR = 3.82 (95% CI; 1.23, 6.55), p = 0.032] to experience fear of death compared with those patients at their early stage of cancer diagnosis. Patients who received care at the public healthcare facilities were more likely [OR = 5.19 (95% CI; 1.80, 15.00), p = 0.002] to experience fear of death relative to those patients who received care at the private healthcare facility. Patients with secondary [OR = 0.18 (95% CI; 0.07, 0.46), p<0.001] and tertiary [OR = 0.08 (95% CI; 0.03, 0.24), p<0.001] level education were less likely to experience fear of death due to their illness compared with those patients with no formal education (Table 3).

**Table 3. Determinants of fear of death among patients due to illness among respondents.**

| Variable | OR [95% CI] | P-value |
|---|---|---|
| Occupation | | |
| Waged worker | 1.00 | |
| Self-Employed | 0.75 [0.32, 1.76] | 0.515 |
| Unemployed | 1.11 [0.19, 6.38] | 0.904 |
| Retired | 0.79 [0.31, 2.00] | 0.615 |
| *Monthly Income (GHC/USD) | | |
| Less than GHC 500 ($89.2) | 1,00 | |
| GHC 500–999 ($89.2–178.4) | 1.06 [0.51, 2.18] | 0.883 |
| GHC 1000–1999 ($178.6–356.9) | 1.15 [0.53, 2.46] | 0.728 |
| GHC 2000–2999 ($357.1–535.5) | 1.12 [0.44, 2.87] | 0.807 |
| GHC 3000–3999 ($535.7–714.1) | 2.03 [0.46, 8.95] | 0.352 |
| GHC 4000+ ($714.3) | 0.48 [0.10, 2.29] | 0.356 |
| Stage of cancer on diagnosis | | |
| Early stage | 1,00 | |
| Middle stage | 1.24 [0.68, 2.27] | 0.489 |
| Advanced stage | 3.82 [1.23, 6.55] | **0.032** |
| Residential Status | | |
| Rural | 1.00 | |
| Urban | 1.27 [0.69, 2.32] | 0.439 |
| Type of Hospital | | |
| Public | 5.19 [1.80, 15.00] | **0.002** |
| Private | 1.00 | |
| Type of cancer | | |
| Breast cancer | 1.00 | |
| Cervical cancer | 0.60 [0.31, 1.17] | 0.133 |
| Highest level of education | | |
| No formal education | 1,00 | |
| Primary | 0.75 [0.29, 1.95] | 0.558 |
| Secondary | 0.18 [0.07, 0.46] | < **0.001** |
| Tertiary | 0.08 [0.03, 0.24] | < **0.001** |

*(Inter-Bank Exchange Rate-Month Average, GHC5.6 = 1 USD, June 2020 (when data was collected)

Source: Authors' data, 2022

Patients who were diagnosed with advanced-stage cancer and had lower chances of cure reported fear of death as a major psychological burden compared with patients who were diagnosed with early-stage cancer and had higher chances of cure. Also, patients who sought treatment at the private hospital reported that they received prompt and better care, easy access to oncologists, and shorter waiting periods at the hospital compared with patients who sought treatment at the public hospitals. With the high quality of care patients received at the private hospital, they hoped for speedy completion of the treatment regimen and resultant cure for their ailment, which reduced the fear of possible death due to cancer. Also, patients with higher education who read a lot about their illness and treatment procedures had their fear of death due to cancer allayed compared with patients with little or no formal education and could not read.

## 3.3 Dietary, physical, and social restrictions

The onset of cancer disrupted the physical and social lives of patients. Apart from the physical discomfort they were experiencing due to the illness, their social activities were halted or slowed. Patients spent several hours at the hospital seeking treatment, they also battled with low energy, physical pain and restrictions, and side effects of treatment. As a result, their participation in family gatherings, church and other religious activities, funerals and parties were severely limited.

About 81.7% of the breast cancer patients studied in the public hospitals, compared to 74.2% of those studied in the private hospital, were unable to attend social events as they used to because of their illness. The majority (82.7%) of cervical cancer patients were also unable to attend social events as they used to. About 88.2% and 83.9% of breast cancer patients studied in the public and private hospitals felt physically restricted because of pain and the need to rest more. Eighty-five percent (85.2%) of cervical cancer patients studied also experienced physical restrictions (see Table 1).

A 52-year-old cervical cancer patient noted:

*"The side effects of the treatment were such that I changed completely. My whole body darkened, and I lost my hair. Because of that I hardly go out. I do not remember the last time I attended a funeral or some social event."*

A 34-year-old cervical cancer patient also stated:

*"Sometimes you get very weak and have pains all over your body, especially after chemotherapy. You are unable to rise from your bed for about 3 days to eat or do anything. Sometimes too you vomit a lot."*

A 41-year-old breast cancer patient stated that she wanted to avoid interrogations from people regarding her illness. As a result, she avoided social gatherings. She highlighted that even though she tried very hard to cover the side effects of treatment by wearing wigs, nail polish and makeup, it was hard to cover her swollen arms. She was afraid someone might notice it and start asking questions. She explained:

*"I have stopped attending social events. I don't want anyone to spot the side effects of treatment and start asking questions. . . besides, where is the energy and money for social activities?"*

This patient also complained about her low energy levels and bad financial situation which did not promote social participation.

A 43-year-old breast cancer patient stated,

*"As part of the side effects of chemotherapy, my right arm got swollen for a long time. I stopped attending social events because people noticed it and started asking questions."*

Furthermore, about 70.4% of the breast cancer patients studied in public hospitals and 83.9% of those studied in the private hospital felt restricted because of dietary changes. Seventy-three percent (73.0%) of the cervical cancer patients studied also reported feeling restricted due to dietary changes (see Table 1). Patients were put on dietary restrictions to help with their recovery and general well-being. Regarding dietary restrictions, a 53-year-old cervical cancer patient indicated:

*"I have been told to stay away from red meat, oily food, and milk. The doctors advise that I should eat more fruits and vegetables, but these are more expensive."*

## 3.4 Perceived stigma and concealment of cancer diagnosis

Stigma and confidentiality regarding cancer played out very evidently among some patients. About 42.5% and 48.4% of breast cancer patients studied in the public and private hospitals, respectively, and 60.5% of the cervical cancer patients studied, reported that they struggled with perceived stigma. As a result, 80.1% and 77.4% of the respondents studied in the public and private hospitals, respectively, hid their cancer diagnosis from people. Similarly, 77.8% of the cervical cancer patients studied also kept their diagnosis a secret from the public (see Table 1).

Most patients indicated that even though people may know in some cases that they were not well, they did not know that it was cancer. Their cancer diagnosis was shared with only close family members and friends who were involved in their treatment process. This included people who accompanied them to the hospital or supported them financially.

A 56-year-old cervical cancer patient explained:

*"My neighbours know that I am sick, but they are not aware that it is cancer. I told them the illness is in my stomach."*

A 54-year-old breast cancer patient who was a head teacher concealed her diagnosis from her colleague teachers. She told her staff that she was going for a meeting whenever she had to go for chemotherapy or radiotherapy sessions. When asked why she chose to be silent about her cancer diagnosis, she indicated:

*"People gossip a lot; I don't want anybody to know my business. And when you have cancer, everybody thinks you are about to die but cancer can be treated. Do I look like someone who is about to die?"*

Fortunately, this patient was very strong and healthy. She was well to do financially, and she also received financial support from the Ghana National Association of Teachers (GNAT) cancer fund. Also, because she was the head teacher, she did not need permission from anyone at work before going to the hospital for her treatment. She could afford to eat well to maintain her health.

While some patients came to the hospital with their spouses and other family members, a 38-year-old breast cancer patient concealed her diagnosis from her husband and relatives. Her

reason was that if her husband got to know about her condition, her mother in-law would automatically be aware and that would mean a broadcast to her entire community. So, she chose to keep the details about her condition to herself. She explained:

> *"Even though my husband knows that there is a problem with my breast, he doesn't know that it is cancer. He is aware that I come to the hospital, but he doesn't know I am doing chemotherapy. I always wear wig in the house to cover my hair loss. I have a very troublesome mother in-law."*

Being a teacher, this patient had support from the GNAT cancer fund, and her husband supported her transportation to the hospital. However, probably because she concealed her disease from her close family members, she lacked the emotional support that most cancer patients had from close relatives.

The regression analysis (Table 4) showed that stigmatization due to illness was significantly associated with patient's occupation and type of cancer. Respondents who were self-employed were more likely [OR = 2.50 (95% CI; 1.10, 5.70), p = 0.030] to be stigmatized due to their illness when compared with waged workers. Cervical cancer patients compared with breast cancer patients were more likely [OR = 1.95, (95% CI; 1.04, 3.65), p = 0.036] to be stigmatized due to their illness.

## 4.0 Discussion

Breast and cervical cancer patients in the study experienced high psychosocial burdens including anxiety, depression and psychological distress, fear of death and loss of loved ones, dietary, physical, and social restrictions, and stigma. Patients with low educational level (no formal education and primary education), who were retired or unemployed, who earned less monthly income and had larger household size, suffered more psychosocial burdens compared to patients with high educational level (secondary and tertiary education) who were gainfully employed, earned higher income, and had smaller household size. This finding supports earlier studies where Cvetković and Nenadović [11] documented depression among 84 breast cancer patients undergoing therapy at the Clinical Centre of Niš in Serbia. Adanu [12] and Gyau [13] recorded moderate and high psychosocial burden among breast cancer patients at the Korle-Bu Teaching Hospital in Ghana. Kugbey, Oppong Asante and Meyer-Weitz [14] reported depression among 205 breast cancer patients who sought treatment at the Korle-Bu Teaching Hospital in Ghana. Alexander et al. [15] found social embarrassment among 378 breast cancer patients seeking treatment at St. John's Research Institute in India. Kugbey, Meyer-Weitz and Oppong Asante [16] highlight that negative psychosocial burden affects the quality of life of cancer patients. Similar experiences were recorded in China where high anxiety was found to have adversely affected the quality of life of cancer patients [17].

Apart from the initial psychosocial burden that being cancer diagnosis brings to patients and their families, financial-related stress was a major source of anxiety, depression, and psychological distress to patients. The exorbitant cost of cancer treatment coupled with the low-income levels of cancer patients in the study explains this phenomenon. Dulcie et al. [18] in their study of the socio- economic effects of cancer on patients in Kenya noted a strong association between developing cancer and loss of income and increased medical expenditure. The authors noted that the high cost of cancer treatment and care subsequently led to a reduction of income of patients which is true for participants in our study. Zafar [19] noted that extreme financial distress produces worse mortality among cancer patients because they may suffer financial-related challenges such as poor health-related quality of life (HRQOL), poor quality

**Table 4. Determinants of stigmatization due to illness among respondents.**

| Variable | OR [95% CI] | P-value |
|---|---|---|
| Occupation | | |
| Waged worker | 1.00 | |
| Self-Employed | 2.50 [1.10, 5.70] | **0.030** |
| Unemployed | 2.54 [0.53, 7.28] | 0.245 |
| Retired | 1.07 [0.42, 2.75] | 0.889 |
| *Monthly Income (GHC/USD) | | |
| Less than GHC 500 ($89.2) | 1.00 | |
| GHC 500–999 ($89.2–178.4) | 0.62 [0.31, 1.22] | 0.165 |
| GHC 1000–1999 ($178.6–356.9) | 0.79 [0.39, 1.60] | 0.508 |
| GHC 2000–2999 ($357.1–535.5) | 1.00 [0.42, 2.40] | 0.995 |
| GHC 3000–3999 ($535.7–714.1) | 2.92 [0.64, 13.35] | 0.167 |
| GHC 4000+ ($714.3) | 0.42 [0.09, 2.09] | 0.291 |
| Stage of cancer on diagnosis | | |
| Early stage | 1.00 | |
| Middle stage | 0.65 [0.37, 1.16] | 0.148 |
| Advanced stage | 1.03 [0.21, 4.92] | 0.974 |
| Residential Status | | |
| Rural | 1.00 | |
| | 0.93 [0.53, 1.64] | 0.803 |
| Type of Hospital | | |
| Public | 1.00 | |
| Private | 1.81 [0.69, 4.73] | 0.226 |
| Type of cancer | | |
| Breast cancer | 1.00 | |
| Cervical cancer | 1.95 [1.04, 3.65] | **0.036** |
| Highest level of education | | |
| No formal education | 1.00 | |
| Primary | 1.12 [0.49, 2.54] | 0.794 |
| Secondary | 0.68 [0.30, 1.53] | 0.354 |
| Tertiary | 0.81 [0.30, 2.16] | 0.670 |

*(Inter-Bank Exchange Rate-Month Average, GHC5.6 = 1 USD, June 2020 (when data was collected)

Source: Authors' data, 2022

of care and poor well-being in general. In a study on the impact of the financial burden of cancer on survivor's quality of life, Fenn et al. [20] also reported that individuals who reported a lot of financial stress also reported poor physical and mental health.

Also, breast and cervical cancer patients in our study experienced dietary, social, and physical restrictions resulting from hospitalisation, body weakness and pain emanating from treatment procedures such as surgery and chemotherapy. The side effects of treatment such as weight loss, darkened palms and feet, and hair loss were visible to others and raised suspicions about patients health status. This deterred our respondents from attending social gathering to avoid unwarranted interrogations. This situation exacerbates feelings of isolation, loneliness, and self-pity among patients.

The fear of perceived stigma was another source of psychosocial burden to breast and cervical cancer patients in our study. Most patients felt the need to keep their diagnosis secret from most people, except a few close family members; principally, persons who accompanied them

for hospital visits and those who provided financial support to them, for purposes of avoiding stigma. This finding corroborates with studies on social stigma of certain diseases like HIV/ AIDS, in Ghana [21, 22]. Stigmatization has the tendency to interfere with the treatment, economic well-being, and quality of life of patients. For instance, studies have shown that HIV/ AIDS patients may not adhere to ART medications or engage in economic activities due to fear of stigma [23].

## 5.0 Conclusion

We studied the psychosocial burden of breast and cervical cancer treatment for patients in three leading cancer hospitals in Ghana. Our study found that the patients studied experienced psychosocial burdens such as anxiety, depression, psychological distress, fear, physical and social restrictions, and stigma, from their illness. Also, respondents had anxieties about their treatment outcome, while some feared that they might die or lose their spouses to divorce or separation due to the stress from their illness. The majority of patients studied also experienced financial-related stress due to the high cost of treatment, especially those who had very little financial resources of their own or support from others. Additionally, most of our respondents were unable to attend social events like they used to do before the onset of cancer. The majority of the respondents also reported going through lots of pain and felt the need to get adequate rest.

There is a need for liaison psychiatrists and health psychologists to assist oncologist to provide psychological support such as free and routine counselling services for the cancer patients and their caregivers. Other studies in Ghana have underscored such need to help persons dealing with ailments such as HIV/AIDS better cope with their illness [9, 17]. This is crucial in alleviating the fears and anxieties of patients. Social support from friends and family is also critical to the well-being of our respondents. There is also need for increased funding from the National Health Insurance for breast and cervical cancer treatment in Ghana. Educational campaigns on mainstream and social media have contributed to increased awareness on breast and cervical cancers in Ghana. The activities of organizations such as Breast Care International and Pink Africa have been critical in increasing public education on cancers in Ghana. There is, therefore, the need for increased educational campaigns to demystify the stigma surrounding cancers in Ghana.

## Acknowledgments

We thank the patients who shared their experiences with us. We are also grateful to the oncology doctors, nurses and other health personnel who helped in making the fieldwork successful.

## Ethics approval

Ethical clearance was obtained from the Ethics Committee for Humanities of the University of Ghana (Ref. No ECH011/19-20), the Korle-Bu Teaching Hospital Ethical Review Board (Ref. No KBTH/MD/G8/20), and the Komfo Anokye Teaching Hospital Review Board (Ref. No KATH-IRB/AP/023/20). Informed written consent was obtained from all study participants before interviews were conducted. Permission was also obtained from research participants to record interviews for the purposes of data analysis and for writing academic-related reports.

## Author Contributions

**Conceptualization:** Patience Gyamenah Okyere Asante, Adobea Yaa Owusu, Joseph Ransford Oppong, Edward Nketiah-Amponsah.

**Data curation:** Patience Gyamenah Okyere Asante, Adobea Yaa Owusu, Kingsley E. Amegah.

**Formal analysis:** Patience Gyamenah Okyere Asante, Kingsley E. Amegah.

**Investigation:** Patience Gyamenah Okyere Asante, Adobea Yaa Owusu, Edward Nketiah-Amponsah.

**Methodology:** Patience Gyamenah Okyere Asante, Adobea Yaa Owusu, Joseph Ransford Oppong, Kingsley E. Amegah, Edward Nketiah-Amponsah.

**Project administration:** Patience Gyamenah Okyere Asante, Adobea Yaa Owusu.

**Resources:** Patience Gyamenah Okyere Asante, Adobea Yaa Owusu, Joseph Ransford Oppong, Kingsley E. Amegah, Edward Nketiah-Amponsah.

**Supervision:** Adobea Yaa Owusu, Joseph Ransford Oppong, Edward Nketiah-Amponsah.

**Validation:** Patience Gyamenah Okyere Asante, Adobea Yaa Owusu, Joseph Ransford Oppong, Kingsley E. Amegah, Edward Nketiah-Amponsah.

**Visualization:** Patience Gyamenah Okyere Asante, Kingsley E. Amegah, Edward Nketiah-Amponsah.

**Writing – original draft:** Patience Gyamenah Okyere Asante, Adobea Yaa Owusu, Kingsley E. Amegah.

**Writing – review & editing:** Adobea Yaa Owusu, Joseph Ransford Oppong, Edward Nketiah-Amponsah.

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
