## [Decision Letter · Decision Letter 0]

28 Mar 2023

PONE-D-23-01781The psychosocial burden of women seeking treatment for Breast and Cervical Cancers in Ghana's major Cancer HospitalsPLOS ONE

Dear Dr. Nketiah-Amponsah,

Thank you for submitting your manuscript to PLOS ONE. After careful consideration, we feel that it has merit but does not fully meet PLOS ONE’s publication criteria as it currently stands. Therefore, we invite you to submit a revised version of the manuscript that addresses the points raised during the review process.

You are please requested to provide adequate information on the following: study population; selection technique used for study participants recruitment; sample size determination; questionnaires used; reason(s) for choice of hospital; and on how themes were formed or built.In addition, please ensure that your results and discussion are presented in an orderly manner, for ease of review.Lastly, you are please advised to attend to all the issues raised by the second reviewer (Reviewer 2)

We look forward to receiving your revised manuscript.

Kind regards,

Olabamiji Abiodun Ajose, MB.BS., M.Sc., MD., FWACP.

Academic Editor

PLOS ONE

“Not applicable”

d) If you did not receive any funding for this study, please state: “The authors received no specific funding for this work.

Additional Editor Comments:

1. Authors are advised to provide adequate information on the following: study population; selection technique used for recruitment of study participants; sample size determination; questionnaires used; reason(s) for choice of hospital; and on how themes were formed.

2. Results and discussion should be presented in an orderly manner, for ease of review.

3. Authors are also advised to attend to all the issues raised by the second reviewer (Reviewer 2)

Reviewers' comments:

Reviewer's Responses to Questions

**Comments to the Author**

1. Is the manuscript technically sound, and do the data support the conclusions?

Reviewer #1: Partly

Reviewer #2: Yes

2. Has the statistical analysis been performed appropriately and rigorously? 

Reviewer #1: No

Reviewer #2: Yes

3. Have the authors made all data underlying the findings in their manuscript fully available?

Reviewer #1: No

Reviewer #2: No

4. Is the manuscript presented in an intelligible fashion and written in standard English?

Reviewer #1: Yes

Reviewer #2: Yes

5. Review Comments to the Author

Reviewer #1: Abstract and introduction: the use of the phrase intangible cost is problematic as the authors tried to measure this cost and used it interchangeably with psycho-social burden.

Methods: this was the most difficult to read. The authors did not give enough information as to study population, how participants were selected, sample size calculation, questionnaires used, how themes were formed and built or even the reason for choice of hospitals. And because of these deficiencies, it was almost impossible to review the results.

Results: the presentation of the results appeared lopsided and it was difficult to follow the flow.

this was the same for the discussion.

Reviewer #2: Psychosocial burden is the outcome variable in this study, yet very little was said about it in the introduction, it was mentioned only in the last paragraph, this is unacceptable, the authors need to introduce this variable to the reader as part of the introduction and not just give it a mention in the last paragraph.

Methodology: Authors should tell us how long the respondents must have been diagnosed before inclusion in the study. It is widely known that duration of illness affect both the level of anxiety and perceived stigma.

This study used both quantitative and qualitative data. It will be good to inform the reader how anxiety, pain and stigma were measured, using a standardized instrument to measure this variables would have been ideal, Howe even if a standard instrument was not used, it is important to inform readers how each of the variables were measured.

Results: Well written and clear

Discussion: Some of the points discussed were out of the purview of the study. For example the authors mentioned risk factors, depression etc which were not really variables studied. Comparing breast and cervical cancer with seizure disorder and leprosy seem far fetched, both epilepsy and leprosy have significant social stigma because symptoms are very obvious to the general population and can hardly be hidden, while for cancers as clearly stated by the authors can remain hidden for a very long time, thus comparing this conditions seems inappropriate. The authors may want to consider comparison with other cancers.

General Remarks: This is a very good study reporting on a situation not often dealt with in the care of the study population.

6. PLOS authors have the option to publish the peer review history of their article (what does this mean?). If published, this will include your full peer review and any attached files.

Reviewer #1: No

Reviewer #2: No

---

## [Author Response · Author response to Decision Letter 0]

25 Jun 2023

Response to Reviewers

The Psychosocial Burden of Women Seeking Treatment for Breast and Cervical Cancers in Ghana's Major Cancer Hospitals

Dear Editor,

We thank you for the opportunity to revise and resubmit our paper for publication consideration. We find the comments very useful and insightful, and they have no doubt contributed to the improvement of our manuscript. The comments as well as the corresponding responses are detailed in the table below. 

Reviewer’s comments

You are please requested to provide adequate information on the following: study population; selection technique used for study participants recruitment; sample size determination; questionnaires used; reason(s) for choice of hospital; and on how themes were formed or built. In addition, please ensure that your results and discussion are presented in an orderly manner, for ease of review.

Authors’ response to comments

The methodology session has been updated to include the study population, issues of sample size, questionnaires, how themes were formed and the reason for the choice of hospitals (pages 4-6). The results and discussion sessions have been updated. Comparison of breast and cervical cancer with leprosy and epilepsy have been taken out (pages 6-19).Also, comments from Reviewer 2 have been addressed.

Reviewer # 1

Abstract and introduction: the use of the phrase intangible cost is problematic as the authors tried to measure this cost and used it interchangeably with psycho-social burden.

Methods: this was the most difficult to read. The authors did not give enough information as to study population, how participants were selected, sample size calculation, questionnaires used, how themes were formed and built or even the reason for choice of hospitals. 

And because of these deficiencies, it was almost impossible to review the results.

Results: the presentation of the results appeared lopsided and it was difficult to follow the flow.

Authors ‘Response to comments

The use of intangible cost has been taken out of the manuscript, and consistently replaced by psycho-social burden.

The methodology session has been updated to include the study population, issues of sample size, questionnaires, how themes were formed and the reason for the choice of hospitals (pages 4-6).

The presentation of results has been updated to enhance easy flow (pages 6-16).

Reviewer #2

Comments

Psychosocial burden is the outcome variable in this study, yet very little was said about it in the introduction, it was mentioned only in the last paragraph, this is unacceptable, the authors need to introduce this variable to the reader as part of the introduction and not just give it a mention in the last paragraph.

Methodology: Authors should tell us how long the respondents must have been diagnosed before inclusion in the study. It is widely known that duration of illness affect both the level of anxiety and perceived stigma.

This study used both quantitative and qualitative data. It will be good to inform the reader how anxiety, pain and stigma were measured, using a standardized instrument to measure this variables would have been ideal, Howe even if a standard instrument was not used, it is important to inform readers how each of the variables were measured.

Results: Well written and clear

Authors ‘response to comments

The abstract and introduction have been updated to cover the psychosocial burden (pages 2-4). 

The minimum duration of cancer diagnosis for patients in the study was 6months. This has been included in the methodology session (page 5).

The methodology session has been updated to include how variables were measured (pages 5-6 ).

Comments 

Some of the points discussed were out of the purview of the study. For example the authors mentioned risk factors, depression etc which were not really variables studied.

Comparing breast and cervical cancer with seizure disorder and leprosy seem far-fetched, both epilepsy and leprosy have significant social stigma because symptoms are very obvious to the general population and can hardly be hidden, while for cancers as clearly stated by the authors can remain hidden for a very long time, thus comparing this conditions seems inappropriate. The authors may want to consider comparison with other cancers.

General Remarks: This is a very good study reporting on a situation not often dealt with in the care of the study population.

Authors ‘response to comments

The discussion session has been updated. Comparison of breast and cervical cancer with leprosy and epilepsy have been taken out (pages 17-19).

I hope our revisions will meet you expectations.

Regards,

Edward Nketiah-Amponsah (PhD)

On-behalf of Co-authors.

---

## [Editor Report · Decision Letter 1]

11 Jul 2023

The psychosocial burden of women seeking treatment for Breast and Cervical Cancers in Ghana's major Cancer Hospitals

PONE-D-23-01781R1

Dear Dr. Nketiah-Amponsah,

We’re pleased to inform you that your manuscript has been judged scientifically suitable for publication and will be formally accepted for publication once it meets all outstanding technical requirements.

Kind regards,

Olabamiji Abiodun Ajose, MB.BS., M.Sc., MD., FWACP.

Academic Editor

PLOS ONE
---

## [Editor Report · Acceptance letter]

11 Aug 2023

PONE-D-23-01781R1 

The Psychosocial Burden of Women Seeking Treatment for Breast and Cervical Cancers in Ghana’s Major Cancer Hospitals 

Dear Dr. Nketiah-Amponsah:

I'm pleased to inform you that your manuscript has been deemed suitable for publication in PLOS ONE. Congratulations! Your manuscript is now with our production department. 

Kind regards, 

on behalf of

Professor Olabamiji Abiodun Ajose 

Academic Editor

PLOS ONE